# Identification of Selected Failures in a Pipe Conveyor's Operation with the Use of the Discrimination Method Based on Continuous Measurement

**Vieroslav Molnár** [1,*] **, Gabriel Fedorko** [2] **, Beáta Stehlíková** [2] **, Peter Michalik** [1] **and Daniel Koštial** [2]

1 Faculty of Manufacturing Technologies with a Seat in Prešov, Technical University of Košice, Bayerova 1, 080 01 Prešov, Slovakia; peter.michalik@tuke.sk
2 Faculty of Mining, Ecology, Process Control and Geotechnologies, Technical University of Košice, Letná 9, 040 01 Košice, Slovakia; gabriel.fedorko@tuke.sk (G.F.); beata.stehlikova@tuke.sk (B.S.); d.kostial@gmail.com (D.K.)
* Correspondence: vieroslav.molnar@tuke.sk; Tel./Fax: +421-55-602-6364

**Abstract:** This paper deals with research on the operational process monitoring of a pipe conveyor for the needs of online diagnostics. The aim of this research is to verify the possibility of identifying the selected pipe conveyor's failures in its straight section during operation (a missing roller in the idler housing, absent material on the conveyor belt) with the use of a discrimination method. This is an attempt to implement digital transformation with the aim of verifying its possibilities and limitations. The basis for discrimination is a continuous measurement and evaluation of measured values of contact forces in certain rollers' positions in the hexagonal idler housing. Within this research, eight different measurement regimes were implemented. The use of the method was verified with simulated data using the trace table. We aimed to create prerequisites for online monitoring, which, based on digital transformation, will be deployed to control a transport system. The measurement was realized with the maximum tension force of 28,000 N. From the measurements, a decision-making algorithm was proposed to identify selected failures in the pipe conveyor operation with the use of the discrimination method. Within the algorithm, classifying criteria were determined, in the range of 57 N ÷ 251 N. The results confirm the method's suitability for its practical assurance of pipe conveyors' failure-free operation, as the failures were always identified sufficiently in advance, thanks to which, in practice, there was no further damage to the diagnosed devices.

**Keywords:** measurement; discrimination method; failure; identification; pipe conveyor





## 1. Introduction

Reliable pipe conveyor operation is every user's basic requirement. Conveyor belts are filled with transported material, and after having been filled, they close into a pipe shape when the belt's edges overlap. One of the options for ensuring the reliable operation of a pipe conveyor is the accurate identification of the pipe conveyor's failures during its operation. Nowadays, during the operation of pipe conveyors, excessive twisting in the closed conveyor belt occurs, significant changes occur in the contact forces on rollers located in the hexagonal idler housings due to a missing roller, and damage to or rupture of the conveyor belt occurs, along with a notable change in the working temperature. Further research and development in this area are aimed at online diagnostics for the mentioned effects.

Due to uneven loads, an uneven direction in bends, or otherwise badly set rollers in the idler housings, belts in the pipe conveyors can rotate around their own axis. The rollers used to maintain the closed pipe shape of the belt in the correct position cause damage to the pipe belt [1]. Early detection of unwanted twisting in the conveyor belt will limit, prevent, or minimize damage to the conveyor system and the loss of transported

material [2]. The company FLSmidth has created a cost-effective and reliable solution for transporting material using a pipe conveyor, which continuously online monitors twisting in the pipe belt using pairs of light detection sensors and ranging technology (LIDAR) [3]. Based on the detection of excessive pipe belt twisting, with the help of a pair of adjustable idler housings, their position and the rollers are automatically adjusted during operation, thereby correcting the twist. A similar principle was also used by Patil et al. [1], who designed the Industrial Pipe Conveyor Belt Orientation Monitoring System. It consists of a microcontroller connected to LIDAR sensors and a Wi-Fi module. The design of the monitoring system can easily monitor the pipe conveyor belt position from a considerable distance and shut down the conveyor if the level of twisting is exceeded. A pipe conveyor monitoring system was also developed by VVV Most [4]. The device is designed to measure the degree of deflection and deformation in the pipe conveyor belt under operating conditions. The design of the monitoring system allows tracking in most pipe conveyor positions. The device evaluates the location of RFID tags and magnetic markers when passing them around detectors and RFID reader antennas, and magnetic markers also determine the vectors of their magnetic moments. Furthermore, a larger number of detectors allows one to ignore time changes in the magnitude of the mark magnetization. The control system monitors the passages of the said magnetic marker points, and from the values obtained with the detectors, determines the transverse position and spatial orientation of the vulcanized marks, and subsequently the degree of folding and deformation in the conveyor belt. The Pipe Conveyor Health Monitoring System [5] uses laser scanning technology to collect pipe conveyor operation data. Using the real-time data, it detects belt swing, the belt overlap position, cuts/tears/breaks in the conveyor belt, and mechanical joints. The state control algorithm generates alerts in the case of any deviation from the set level on the computer monitor. The system can be configured to stop the conveyor belt.

The good technical condition of rollers guarantees minimal costs for the conveyor's operation. Monitoring contact forces on rollers is currently just beginning to be applied in pipe conveyors. It is related to ongoing research in laboratories using test stands [6–10]. Using a test stand, Kulinowski et al. [11] studied the rotational resistance of the roller under operating load. The authors found a linear increase in the rotational resistance of the roller with an increased radial load of the roller as well as the roller's rotational speed. Peruń and Opasiak [12] carried out vibration measurements of the roller shell installed on a test stand. By comparing the results from analyses of new and worn rollers at further stages in their studies, the authors determined limit levels for vibroacoustic phenomena, after exceeding which, the roller needs to be replaced. Klimenda et al. [13] measured the noise and vibration in a rubber conveyor belt when not loaded. Their research results make it clear that the noise and vibration in the conveyor belt structure were reduced using rollers made of precision tubes. Kuster et al. [14] used driven rollers on a test stand, which reduced the load on a conventional actuator of the pipe conveyors. Thus, the actuator can be smaller, which leads to energy and cost savings.

Another crucial factor that affects pipe conveyors' reliable operation is the dynamics of the conveyor belts (dynamic characteristics of the belt conveyor). These are determined by the belt's properties. Several authors are engaged in this area. Li et al. [15] investigated a belt's frequency and dynamic stress using digital simulation. The results showed that the belt conveyor's frequency decreases when its throughput increases. Hou and Meng [16] carried out experiments to determine a conveyor belt's dynamic properties. They measured the dynamic modulus of elasticity, viscous damping, and rheological constants of the belt. They investigated longitudinal vibrations, natural vibration frequency in the transverse direction, and response to impulse actuating. They found that the speed of stress wave propagation increased with a tensile load, and the tensile load was the main factor influencing the longitudinal vibration. Kusumaningtyas et al. [17] investigated the axial elastic response of the conveyor belt with the help of performed simulations when starting unloaded belt conveyors with various numbers of actuators. For each simulation case, they

assessed linear starting. The simulations revealed that using multiple actuators reduced the maximum stress on the conveyor belt in both non-stationary and stationary conditions. Halepoto and Khaskheli [18] proposed an energy-efficient model for a conveyor system that controls several actuator units, thereby increasing its efficiency. Khodusov et al. [19] dealt with the causes of dynamic stress during the movement of a conveyor belt with material load. They defined the main factors of dynamic resistance in the transport system and found that the speed of failure propagation depends on the level of material load on the conveyor belt. Bortnowski et al. [20] analyzed belt vibrations, which provide information on the conveyor's operation. They measured the frequency of a blet's transverse vibrations along the entire length of its route without the presence of material on the conveyor belt and with the belt loaded using a specially made device located on the conveyor belt.

This paper deals with research on issues for online diagnostic needs during a pipe conveyor's operation. This research aims to verify the possibility of identifying selected pipe conveyor failures in its straight section during operation (a missing roller in the idler housing, absent material on the conveyor belt) with the use of a discrimination method. Discrimination, in this case, is an estimate that the device is in a failure-free state or an estimate for the type of device failure, based on continuous measurement and evaluation of measured contact forces for given positions of the rollers in the hexagonal idler housing.

Until now, most of the presented papers have dealt with the detection of various failures in continuous transport systems, for example, [21–25]. Their common feature is the use of specific methods to identify various types of failures in continuous transport systems. However, within the research they present, their effort is to identify failures using quantifiable measurable parameters, i.e., so that the undesirable state of a continuous transport system can be identified using digital measurements. The authors do not attempt to verify individual methods (such as ultrasonic detection, temperature detection, and the like), but based on a change in tension values, they attempt to directly identify the emergence and existence of an undesirable condition and to digitally transform the whole transport equipment to monitor its operation.

This research aimed to find a method that, based on experimental measurements, will enable the classification of continuous transport system operation (pipe conveyor) and the identification of undesirable conditions. This research considered:

- Standard operation (failure-free condition);
- Failure in the idler housing (e.g., a missing roller);
- Absence of material on the conveyor belt.

Considering how the measured data are evaluated, the method for failure identification is called discrimination, and the proposed method is discriminatory, as this term accurately denotes it. It is a process or procedure by which operational states are evaluated. Based on their digital transformation, they are then classified into an appropriate category, according to measured characteristics.

This research aims to enhance the digital transformation of continuous transport systems. This is a complex and demanding process; however, its realization is inevitable due to intra-logistic system transformation, in line with Industry 4.0 requirements. Digital transformation is also a must in terms of logistic process automation, thereby increasing effectiveness and reducing operating costs. As part of the presented research, the authors tried to focus on basic starting points and the verification of a suitable method that will lead to further research.

## 2. Materials and Methods

All the measurements of contact forces (CFs) on an idler housing's rollers [26] were performed on a specially developed test stand [27], on which 3 hexagonal idler housings with a closed conveyor belt were placed. A view of the test stand is shown in Figure 1, and its basic features are listed in Table 1. The parameters of the used conveyor belt are listed in Table 2.

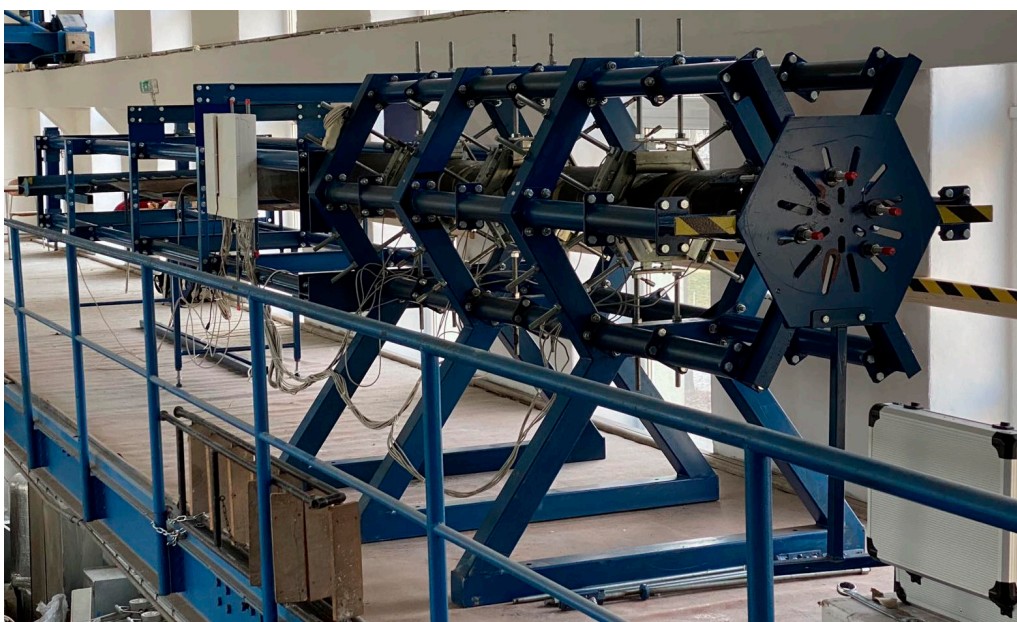

**Figure 1.** A view of the test stand.

**Table 1.** Basic characteristics of the test stand.

| | |
|---|---|
| Length | 9242 mm |
| Width | 1620 mm |
| Height | 1880 mm |
| Closed belt's diameter—maximum | 325 mm |
| Closed belt's diameter—minimum | 110 mm |
| Distance between idler housings—maximum | 2000 mm |
| Distance between idler housings—minimum | 550 mm |
| Weight | 1072 kg |

**Table 2.** Basic characteristics of the conveyor belt.

| Type of Conveyor Belt | Polyester Rubber Textile Conveyor Belt EP500/3 HP 5 + 3 D |
|---|---|
| Strength of the belt | 500 N mm$^{-1}$ |
| Thickness of the conveyor belt | 20 mm |
| Width of the conveyor belt | 800 mm |
| Diameter of the pipe shaped conveyor belt | 200 mm |
| Distance between the idler housings | 1000 mm |
| Diameter of the idler rolls | 60 mm |

The selected designation of measurements corresponds with the designation used in experimental measurements. As part of this research, eight measurement modes were implemented:

- Measurement of contact forces (CFs) with material simulating a failure-free state (designation m200);
- Six measurement modes in which the model was filled with a material with a single roller always missing at different positions ID7 ÷ ID12 (designation m207, m208, m209, m210, m211, and m212, in terms of Figure 1);
- Measurement of contact forces without material, simulating the loss of material from the conveyor (designation TWOMA0).

The maximum value of the conveyor belt's tension force was 28,000 N. The contact forces (CFs) were measured only on the central idler housing with positions of the measuring rollers as designated in Figure 2.

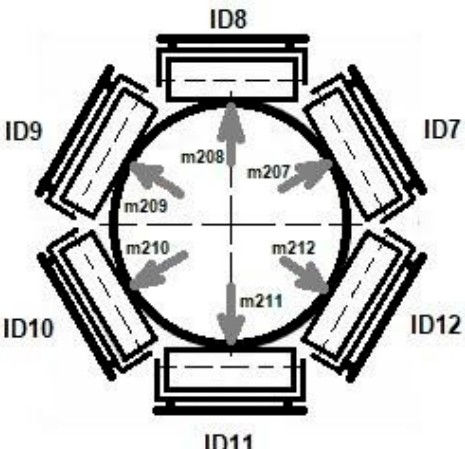

**Figure 2.** Positions of the measuring rollers ID7 ÷ ID12 and designation of the measurement type.

To determine the discrimination procedure and parameters, a cluster analysis and Euclidean distance decision tree, logical considerations, and original, unprocessed measurement records were used.

### 2.1. Cluster Analysis and Euclidean Distance

Cluster analysis is one of the statistical methods for organizing items into groups, which are named clusters, by computing how closely associated they are. Cluster analysis is realized with data matrices where the relation between variables has not been mentioned. The goal of cluster analysis is to find similar groups of data objects. The similarity between each pair of data objects means some global measure over the whole set of the characteristics of the data objects. Cluster analysis belongs to unsupervised learning algorithms. Before running the model, it is not known how many clusters exist in the data. It gives information about if and where data object associations exist in the data. It does not answer the question of what the cause of associations is. The most common task that uses cluster analysis is classification. Here, data objects are separated into groups so that each data object is more similar to the other data objects in its group than it is to data objects outside the group. It is advisable to use a software tool with this functionality for building classification and regression trees [28].

Euclidean distance is one of the metrics used to determine the distance of objects. Its value in multidimensional space is equal to the absolute value of the distance between the two points (1).

$$m_e\left(\vec{a}, \vec{b}\right) = \sqrt{\sum_{i=1}^{n}(a_i - b_i)^2} \tag{1}$$

where $\vec{a}$ and $\vec{b}$ are vectors with the same number of elements.

### 2.2. Decision Tree

Nowadays, decision trees are extremely popular among researchers that find applications in machine learning methods. They are used for both classification and regression tasks. A decision tree is a hierarchical, multi-level, binary decision system, in which the fulfillment/non-fulfillment (if/else) of the decision criteria or conditions is evaluated step by step until an accepted class or a solution is obtained. The decision-making process proceeds from the tree root, gradually through the individual nodes. Decision-making criteria in individual nodes are mostly sorted according to informational importance. The largest weight criterion allowing the best input data separation into two binary classes (yes/no) becomes the root of the tree. Other nodes are successively formed using the remaining criteria with smaller weights, where each node generates two binary solutions.

Before implementing the model, the pros and cons of a decision tree were considered. The advantages of a decision tree, which supported our decision, are:

- Easy to understand and interpret as it is a visual representation of the decision-making process. It can be useful to see the progression of how each factor affects the decision.
- It can use both categorical and numerical data.
- It can handle missing values and outliers in a data set. This makes decision trees a robust tool that can handle real-world data.
- It can be used for classification and regression problems.
- It can help identify important relations in the data.

The disadvantages of a decision tree, which we considered, are:

- The decision model can be too complex and overfit.
- Decision trees can be sensitive to small changes in the data.
- Large decision trees can be difficult to interpret.

## 3. Results

The procedure for identifying selected failures in pipe conveyor operation with the use of the discrimination method is shown in Figure 3.

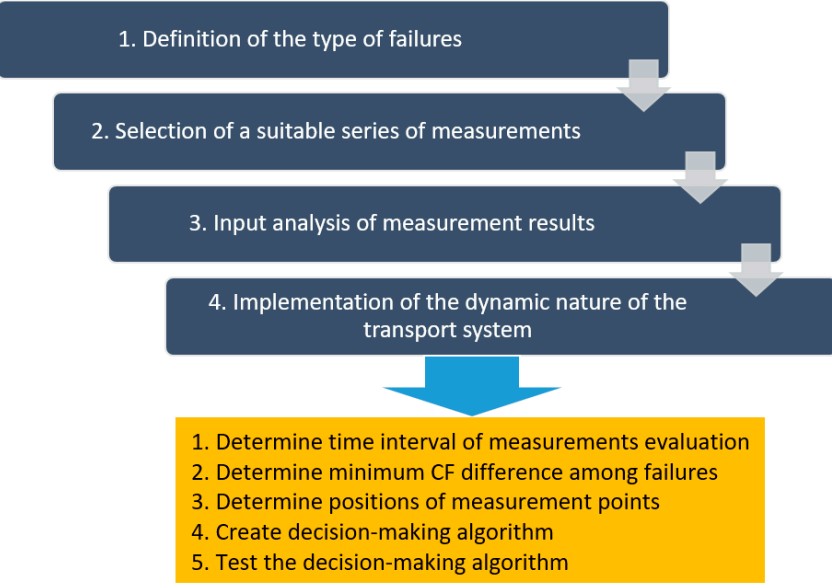

**Figure 3.** Procedure for identifying selected failures in the operation of a pipe conveyor using the discrimination method.

In step 1, for the identification of selected failures in the pipe conveyor's operation using the discrimination method (Figure 3), the following monitored types of failures were defined:

(a)   A missing roller ID7 ÷ ID12;
(b)   Absent material on the conveyor belt.

The identification of selected failures in the pipe conveyor's operation using the discrimination method continued in step 2 by selecting a suitable series of measurements. The analyses were conducted for a tension force of 28,000 N. For the purposes of this research, the following measurements were performed:

(a)   Failure-free operation of the conveyor belt with material (designation m200);
(b)   With a missing roller at position ID7 (designation m207), a missing roller at position ID8 (designation m208), . . . , up to a missing roller at position ID12 (designation m212);
(c)   With material absence on the conveyor belt (designation TWOMA0).

In step 3, the identification of the selected failures in the pipe conveyor's operation, using the discrimination method (Figure 3), was followed by an input analysis of the measurement results to define the research task in detail. To display the similarity and

layout of contact forces (CFs) for individual set measurements, the average values of contact forces (CFs) were calculated as the average of twenty consecutively measured data points during 30 s from the moment when the tension force in the conveyor belt reached the value of 28,000 N. If the failure was due to a missing roller, the average value of the contact force (CF) at its position was zero. The results for the measurements of contact forces (CFs) were obtained when the conveyor belt was not moving. The analysis of measurement results consisted of verifying the possibility of unambiguous failure identification. The following reasoning and Euclidean distance calculation were used for individual measurements listed in Table 3.

**Table 3.** Euclidean distance for the individual experimental measurements (N).

| Measurement Mode | m200 | m207 | m208 | m209 | m210 | m211 | m212 |
|---|---|---|---|---|---|---|---|
| m207 | 2.39 | | | | | | |
| m208 | 2.84 | 4.06 | | | | | |
| m209 | 2.80 | 3.61 | 4.32 | | | | |
| m210 | 3.36 | 3.51 | 4.86 | 3.70 | | | |
| m211 | 3.69 | 2.82 | 5.26 | 3.35 | 3.51 | | |
| m212 | 3.72 | 3.41 | 4.39 | 3.90 | 2.82 | 3.68 | |
| TWOMA0 | 0.78 | 2.16 | 2.58 | 2.92 | 2.93 | 3.53 | 3.07 |

Note: In this special case, Euclidean distance is given in physical units, as each attribute by which the objects (measurements) are defined is in (N) units.

Each measurement was considered an object with attributes of the average values of contact forces (CFs) at all six positions of rollers ID7 ÷ ID12 (Figure 2). These values correspond to the difference in average values of contact forces (CFs). The smallest is the Euclidean distance of objects in the failure-free operation of the conveyor belt m200, and the operation of the conveyor belt without material TWOMA0. There is a minimum difference in Euclidian distance (0.78 N ÷ 3.72 N), highlighted in purple, during failure-free operation and a failure (second column in Table 3). Such a minor increase in contact forces (CFs) can occur at any time during the failure-free operation, and, therefore, the failure cannot be precisely identified from the given Euclidean distance. It is clear from the results that for precise failure identification, the measurement points of contact forces (CFs) need to be determined, at which the change in contact forces (CFs) for individual failures is significantly larger. The Euclidean distance values are also shown as a dendrogram in Figure 4. The graphic display makes it clear that a different method must be chosen to unambiguously identify failures.

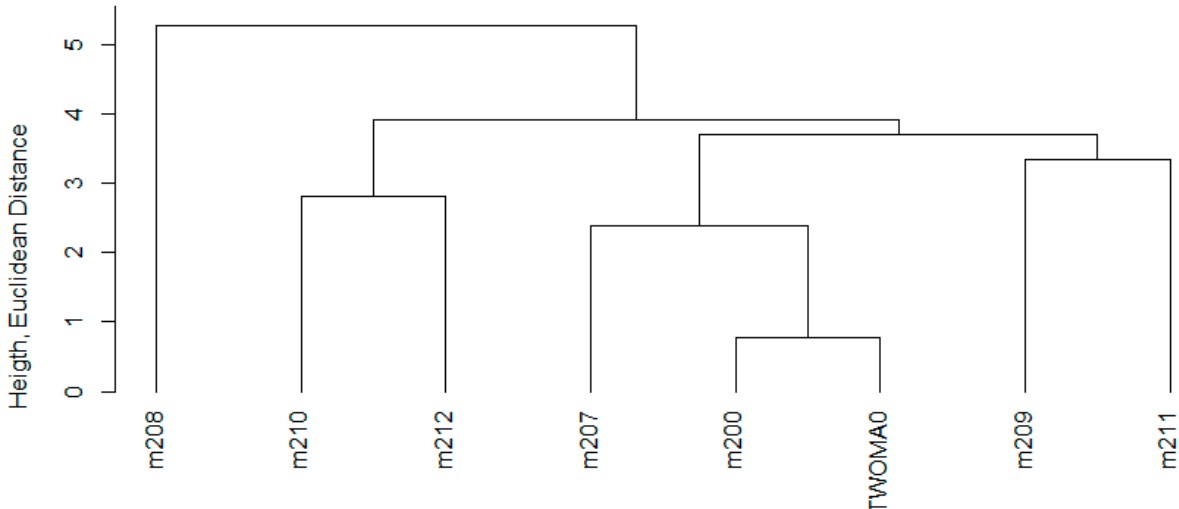

**Figure 4.** Dendrogram.

To display average values of the contact forces (CFs) for individual measurements, a simple radar graph was used, as shown in Figure 5, in which the average values are the contact forces (CFs) at roller positions ID7 ÷ ID12 shown in the grid. Each measurement is displayed as a series of data.

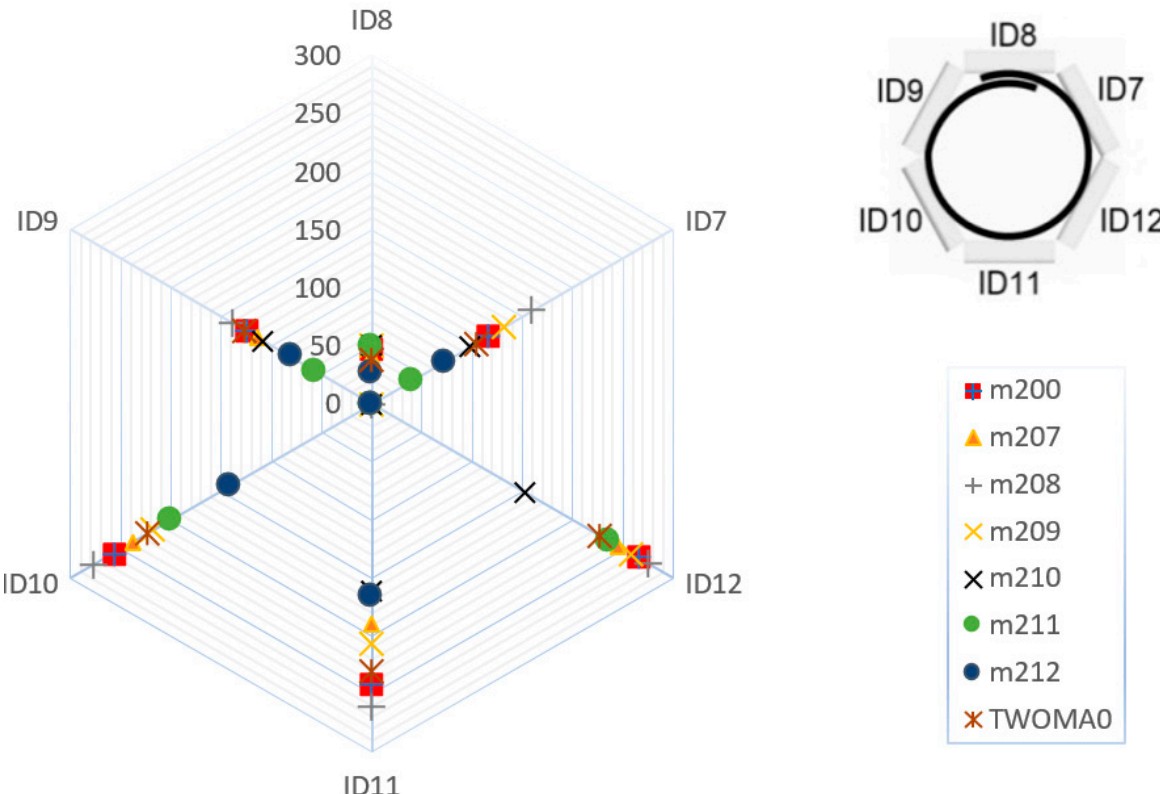

**Figure 5.** Average values of the contact forces (CFs) at roller positions ID7 ÷ ID12.

Figure 4 and the Euclidean distance calculation make it clear that to identify the failure correctly and unambiguously, the focus must be on:

- Positions that are most affected by the failure (differences in contact forces (CFs) for individual failures and a failure-free state are large enough);
- Determining the number of measured positions so that the classification is valid and dependable.

In step 4, for the identification of selected failures in the pipe conveyor's operation using the discrimination method, the dynamic nature of the actual pipe conveyor was considered. Due to its dynamic nature (dynamics of the transportation process, variance in measurement results, and flexibility of the conveyor belt material), an estimate was needed for the time interval for evaluating continuous contact force (CF) measurements. Determining the minimum value of the difference in contact forces (CFs) between a failure-free operation and a failure, as well as the difference in the value of contact forces (CFs) between individual failures, relates to the dynamic nature of the transport system. Previous measurements show that after setting the tension force in the conveyor belt, the conveyor belt becomes relaxed, and the tension force as well as the contact forces (CFs) gradually stabilize. However, this does not happen in real conditions since the system is dynamic, and changes in contact forces (CFs) and the tension force in the conveyor belt are triggered by the process of transport. Therefore, it is necessary to use the results of measurements that record contact force (CF) dynamics and the tension force during relaxation, and based on them, estimate the required data (time interval for continuous measurements evaluation, determining the minimum value of a difference in contact forces (CFs) between the failure-

free operation and a failure, as well as the difference in the value of contact forces (CFs) between individual failures).

Using the discrimination method, the above four steps for classifying selected failures in an idler housing and a conveyor belt, in the straight sections of the pipe conveyor, resulted in a formulation of partial research tasks:

1.  Determine the time interval for measurement evaluation;
2.  Determine the minimum value of the difference in contact forces (CFs) between a failure-free operation and a failure, as well as the difference in contact forces (CFs) between individual failures;
3.  Determine the positions of measuring points to measure contact forces (CFs) on rollers, so that it is possible to clearly identify the defined failures, based on the measured values of contact forces (CFs);
4.  Create a decision-making algorithm to determine a failure-free state and a failure;
5.  Verify the validity and reliability of the decision-making algorithm using simulated data.

Under the procedure mentioned in this Section, experimental research was carried out to verify the hypothesis that using data obtained continuously by measuring the selected operational parameters of a pipe conveyor, with the use of a discrimination method, it is possible to identify undesirable situations that jeopardize the pipe conveyor's operation and reduce its reliability.

### 3.1. Determination of a Time Interval for Measurement Evaluation

To determine conditions of the time interval for the evaluation of contact forces (CFs), and for the minimum acceptable difference in contact forces (CFs), the data from the m200 measurement were used, which were obtained when the moment of tensioning termination was identified (the maximum value for tensioning forces in the transport belt of 28,000 N was achieved). The subtracted instant values after 5, 10, 15, 20, 30, 60, and 90 s are listed in Table 4. After 90 s, the condition was considered stable.

**Table 4.** Contact forces (CFs) measured after 5, 10, ..., and 90 s from tensioning termination.

| Time from Tensioning Termination (s) | CF (N) | | | | | |
|---|---|---|---|---|---|---|
| | ID7 | ID8 | ID9 | ID10 | ID11 | ID12 |
| 5 | 73.6 | 46.1 | 155.4 | 242.8 | 243.3 | 215.1 |
| 10 | 73.4 | 47.1 | 154.5 | 243.1 | 236.6 | 216.2 |
| 15 | 72.8 | 46.5 | 153.0 | 241.4 | 235.5 | 217.0 |
| 20 | 71.7 | 46.3 | 154.1 | 241.0 | 233.6 | 217.8 |
| 30 | 70.5 | 45.6 | 152.5 | 241.1 | 230.3 | 220.4 |
| 60 | 72.1 | 46.2 | 151.9 | 238.1 | 223.5 | 222.1 |
| 90 | 71.8 | 46.2 | 151.8 | 237.4 | 219.2 | 224.4 |

In Table 5, to record the dynamics of relaxation, differences in contact forces (CFs) were calculated after 10, 15, ..., and 90 s and 5 s from tensioning termination ($\text{ID7CF}_{10s} - \text{ID7CF}_{5s} = 0.2$ N, $\text{ID7CF}_{15s} - \text{ID7CF}_{5s} = 0.8$ N, ..., and $\text{ID12CF}_{90s} - \text{ID12CF}_{5s} = -9.3$ N).

**Table 5.** Calculated differences in contact forces (CFs) after 10, 15, ..., and 90 s from contact forces (CFs) 5 s after tensioning termination.

| Time from Tensioning Termination (s) | CF Difference (N) | | | | | |
|---|---|---|---|---|---|---|
| | ID7 | ID8 | ID9 | ID10 | ID11 | ID12 |
| 10 | 0.2 | −1.0 | 0.9 | −0.4 | 6.7 | −1.1 |
| 15 | 0.8 | −0.3 | 2.4 | 1.3 | 7.8 | −2.0 |
| 20 | 2.0 | −0.2 | 1.3 | 1.8 | 9.8 | −2.8 |
| 30 | 3.1 | 0.5 | 2.9 | 1.6 | 13.0 | −5.3 |
| 60 | 1.5 | −0.1 | 3.5 | 4.7 | 19.8 | −7.0 |
| 90 | 1.9 | −0.1 | 3.8 | 5.4 | 24.2 | −9.3 |

The largest difference in contact forces (CFs) for a steady state was at position $\text{ID11CF}_{90s} - \text{ID11CF}_{5s} = 24.2$ N, highlighted in yellow, i.e., 85 s from tensioning termination. The same calculation was performed (Table 6) to determine differences in each contact force (CF) at the position after 20, 30, ..., and 90 s and 15 s from the tensioning termination ($\text{ID7CF}_{20s} - {}_{\text{ID7CF15s}} = 1.2$ N, $\text{ID7CF}_{30s} - \text{ID7CF}_{15s} = 2.4$ N, ..., and $\text{ID12CF}_{90s} - \text{ID12CF}_{15s} = -7.4$ N).

The largest difference in contact forces (CFs) for a steady state was at position $\text{ID11CF}_{90s} - \text{ID11CF}_{15s} = 16.3$ N, highlighted in yellow, i.e., 75 s from tensioning termination. Based on these analyses, an evaluation interval of 15 s was determined for directly measured contact forces (CFs) as follows: If the measurements identify a change in contact forces (CFs), the evaluation of whether it is a failure and of what kind will be carried out during the measurement that follows after the next 15 s.

**Table 6.** Calculated differences in contact forces (CFs) at 20, 30, ..., and 90 s and 15 s from the tensioning termination.

| Time from Tensioning Termination (s) | CF Difference (N) | | | | | |
|:---:|:---:|:---:|:---:|:---:|:---:|:---:|
| | ID7 | ID8 | ID9 | ID10 | ID11 | ID12 |
| 20 | 1.2 | 0.2 | −1.0 | 0.4 | 2.0 | −0.8 |
| 30 | 2.4 | 0.9 | 0.5 | 0.3 | 5.2 | −3.4 |
| 60 | 0.7 | 0.2 | 1.1 | 3.3 | 12.0 | −5.0 |
| 90 | 1.1 | 0.3 | 1.2 | 4.0 | 16.3 | −7.4 |

*3.2. Determination of Minimal Value Differences in Contact Forces (CFs) between a Failure-Free Operation and a Failure, and the Difference in Contact Forces (CFs) between Individual Failures*

Considering the fact that the largest difference in contact forces (CFs) for a steady condition was at position $\text{ID11CF}_{90s} - \text{ID11CF}_{15s} = 16.3$ N (Table 6), a minimum acceptable difference in contact forces (CFs) for failures identification was defined as 20 N.

*3.3. Determination of Measuring Positions to Measure Contact Forces (CFs) on Rollers So That Defined Failures Can Be Identified on the Basis of Measured Contact Force (CF) Values*

We set the following criteria for measuring roller positions that are suitable for ongoing measurements to classify failures:

- A minimum number of differences in contact forces (CFs) (between a failure-free state and failures and also between failures) smaller than 20 N.
- The combination of roller positions must ensure that, for every difference (between a failure-free state and a failure and between failures), there is a roller position for which the difference is more than 20 N.

To meet these criteria, the following procedure was used:

1. Calculate the mutual differences between the average contact forces (CFs) at each roller's position in a failure-free state, and for each failure, in accordance with the following procedure:

   $\text{ID7CF}_{m200} - \text{ID7CF}_{m207}$ (is empty), $\text{ID7CF}_{m200} - \text{ID7CF}_{m208} = 44.4$ N, ..., and $\text{ID7CF}_{m212} - \text{ID7CF}_{\text{TWOMA0}} = 32.0$ N, as listed in Table 7. Mutual differences are given only once. For the type of failure with a missing roller at the measuring point, the difference in measurement is not considered, and the cell is empty. Differences that are unaccepted (their value is smaller than 20N) are displayed in red.

2. Determine the number of unsatisfactory measurement differences (smaller than 20 N) for each roller position. These values are the red cells in Table 7. The number of unsatisfactory differences was obtained for each roller position (Table 8).

**Table 7.** Mutual differences between contact forces (CFs) measured at the ID7 roller position for a failure-free condition and failures.

| Failure m200 | Mutual Differences in Contact Forces (CFs) (N) | | | | | | |
|---|---|---|---|---|---|---|---|
| | m200 | m207 | m208 | m209 | m210 | m211 | m212 |
| m207 | | | | | | | |
| m208 | 44.4 | | | | | | |
| m209 | 17.5 | | 27.0 | | | | |
| m210 | 18.8 | | 63.2 | 36.3 | | | |
| m211 | 74.5 | | 118.9 | 92.0 | 55.7 | | |
| m212 | 42.0 | | 86.4 | 59.4 | 23.2 | 32.5 | |
| TWOMA0 | 9.9 | | 54.4 | 27.4 | 8.9 | 64.6 | 32.0 |

**Table 8.** The number of unsatisfactory measurement differences for the roller position.

| Roller Position | ID7 | ID8 | ID9 | ID10 | ID11 | ID12 |
|---|---|---|---|---|---|---|
| number of unsatisfactory differences | 4 | 17 | 8 | 4 | 4 | 8 |

Table 8 makes it clear that the best positions to classify a failure are ID7, ID10, and ID11, where the smallest number of unsatisfactory differences is equal to four.

3. Select the roller position combinations that meet the criterion (for each difference, the roller's position exists with a difference larger than 20 N). The validity of this criterium was verified for roller positions chosen in the previous procedure step, i.e., ID7, ID10, and ID11. Table 9 presents the positions for given pairs of the states (without a failure, with a failure), which, based on the differences in contact forces (CFs) on the roller, cannot be distinguished from each other. These are the red cells in Table 9.

**Table 9.** Pairs of states (without a failure and a failure) that cannot be distinguished from each other.

| Roller Position | ID7 | ID10 | ID11 |
|---|---|---|---|
| not met | $ID7CF_{m200}$–$ID7CF_{m209}$ | $ID10CF_{m200}$–$ID10CF_{m207}$ | $ID11CF_{m200}$–$ID11CF_{m208}$ |
| not met | $ID7CF_{m200}$–$ID7CF_{m210}$ | $ID10CF_{m207}$–$ID10CF_{TWOMA0}$ | $ID11CF_{m200}$–$ID11CF_{TWOMA0}$ |
| not met | $ID7CF_{m200}$–$ID7CF_{TWOMA0}$ | $ID10CF_{m209}$–$ID10CF_{m211}$ | $ID11CF_{m207}$–$ID11CF_{m209}$ |
| not met | $ID7CF_{m210}$–$ID7CF_{TWOMA0}$ | $ID10CF_{m209}$–$ID10CF_{TWOMA0}$ | $ID11CF_{m210}$–$ID11CF_{m212}$ |

The pair of states m200 and TWOMA0 that cannot be distinguished from each other, occur (Table 9) at roller positions ID7 and ID11 (red cells). For the ID10 roller position, this pair of states meets the difference in contact force (CF) value, i.e., both criteria are met for the selected combination of ID7, ID10, and ID11.

### 3.4. Decision Algorithm to Determine a Failure-Free State and a Failure

We used a decision tree to obtain an algorithm for the classification of states of individual failures [29–31]. The data were divided into two parts: a training set and a testing set. The training set contained 80% of the data, and the testing set contained 20% of the data. Figure 6 presents an example of a decision tree used to identify-classify a failure: a missing roller at position ID9. For every decision tree, its reliability was calculated based on the number of correctly classified objects (measurements) from the testing data set. The reliability of every decision tree was 100%.

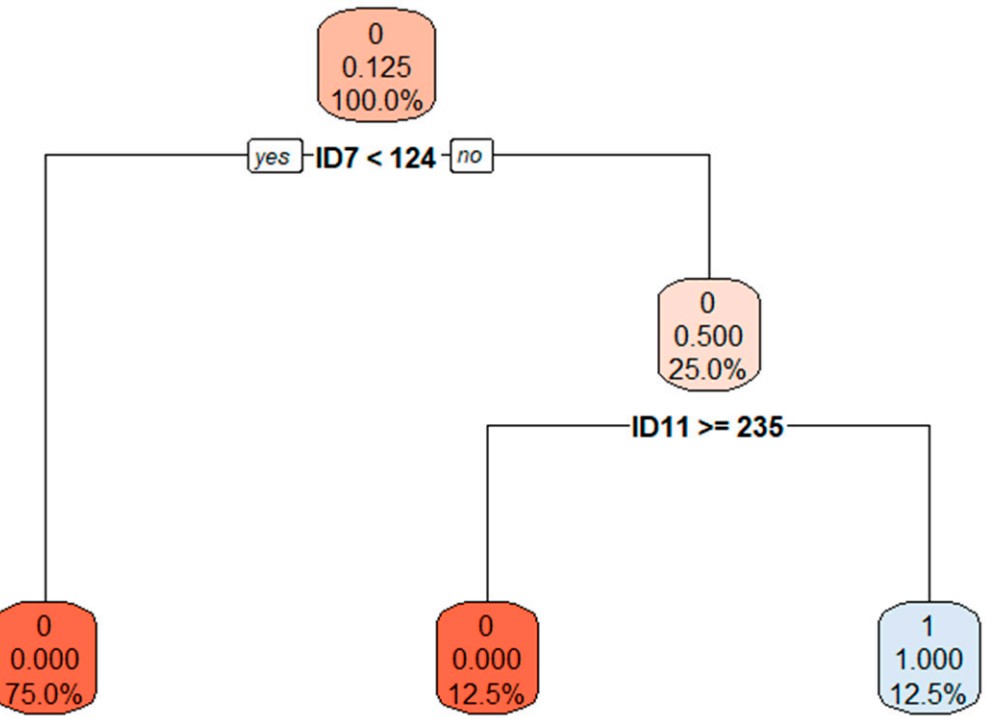

**Figure 6.** The decision tree used to identify a missing roller at position ID9.

The classification criteria for a failure-free condition and a failure in the pipe conveyor operation, using the discrimination method, are listed in Table 10.

**Table 10.** Classification criteria for a failure-free condition and a failure.

|  | ID7 |  |  |  | ID10 |  |  |  | ID11 |  |  |
|---|---|---|---|---|---|---|---|---|---|---|---|
| m200 |  | ID7 < 138 | ∩ | ID10 ≥ 248 |  | ∩ | ID11 ≥ 236 | ∩ | ID11 < 251 |  |  |
| m207 |  |  |  | ID10 ≥ 231 |  | ∩ |  | ID11 < 215 |  |  |  |
| m208 | ID7 ≥ 146 |  | ∩ |  |  |  | ID11 ≥ 251 |  |  |  |  |
| m209 | ID7 ≥ 124 |  | ∩ |  |  |  | ID11 < 235 |  |  |  |  |
| m210 | ID7 ≥ 85 | ∩ | ID7 < 101 | ∩ |  |  | ID11 ≥ 79.6 | ∩ | ID11 < 163 |  |  |
| m211 |  | ID7 < 57 | ∩ | ID10 < 220 |  |  |  |  |  |  |  |
| m212 |  |  |  | ID10 < 170 | ∩ | ID11 ≥ 163 |  |  |  |  |  |
| TWOMA0 | ID7 ≥ 101 | ∩ | ID7 < 110 | ∩ | ID10 ≥ 221 | ∩ | ID10 < 231 | ∩ | ID11 ≥ 220 | ∩ | ID11 < 236 |

Figure 7 presents the decision algorithm used to identify selected failures in the pipe conveyor operation with the use of the discrimination method as in Table 10. Testing of this algorithm can be found in Section 4, which discusses the decision algorithm's validity and reliability using simulated data.

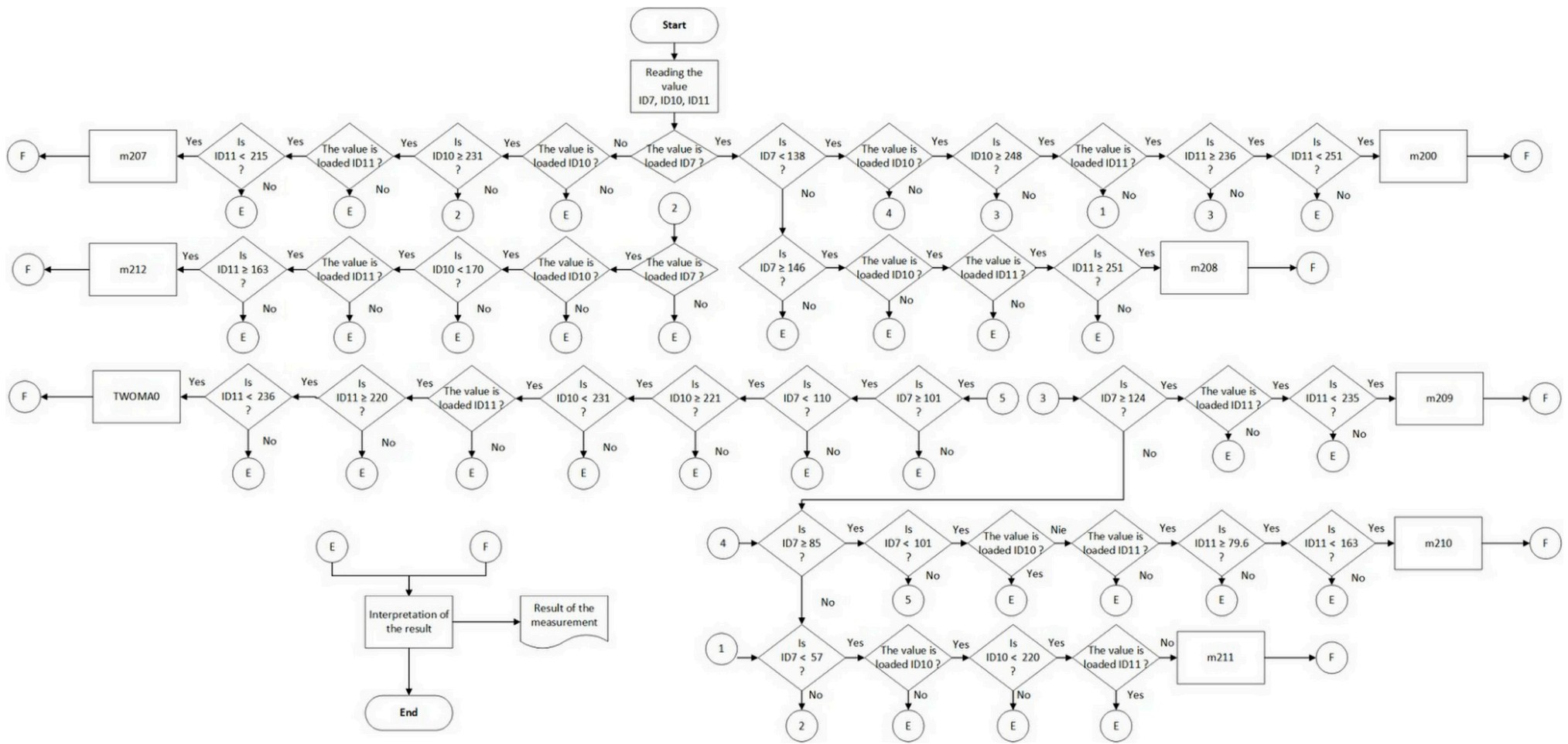

**Figure 7.** Decision-making algorithm used to identify selected failures in the pipe conveyor operation with the use of the discrimination method.



## 4. Discussion of the Decision Algorithm's Validity and Reliability Using Simulated Data

The trace table method was used to verify the adequacy of the decision-making algorithm. Table 11 presents the simulation inputs and outputs, using the trace table method for all types of failures. The individual consecutive simulated and measured values of contact forces (CFs) are listed in Table 11 for each step. The time interval between individual steps was 15 s.

**Table 11.** Example of simulation inputs and outputs using the trace table method for all types of failures.

| Step | ID7 | ID10 | ID11 | Output | | Step | ID7 | ID10 | ID11 | Output |
| --- | --- | --- | --- | --- | --- | --- | --- | --- | --- | --- |
| | Input | | | | | | Input | | | |
| Step 1 | 115 | 257 | 241 | m200 | | Step 1 | 113 | 252 | 249 | m200 |
| Step 2 | 123 | 235 | 252 | E | | Step 2 | 123 | 235 | 252 | E |
| Step 3 | 123 | 235 | 252 | E | | Step 3 | 123 | 235 | 252 | E |
| Step 4 | 112 | 254 | 246 | m200 | | Step 4 | 113 | 257 | 243 | m200 |
| Step 5 | x | 254 | 243 | E | | Step 5 | 159 | 280 | 263 | m208 |
| Step 6 | x | 238 | 191 | m207 | | | | | | |

| step | ID7 | ID10 | ID11 | output | | step | ID7 | ID10 | ID11 | output |
| --- | --- | --- | --- | --- | --- | --- | --- | --- | --- | --- |
| | input | | | | | | input | | | |
| Step 1 | 114 | 255 | 237 | m200 | | Step 1 | 114 | 254 | 241 | m200 |
| Step 2 | 123 | 235 | 252 | E | | Step 2 | 123 | 235 | 252 | E |
| Step 3 | 123 | 235 | 252 | E | | Step 3 | 123 | 235 | 252 | E |
| Step 4 | 115 | 258 | 239 | m200 | | Step 4 | 112 | 257 | 239 | m200 |
| Step 5 | 140 | 215 | 200 | E | | Step 5 | 82 | x | 170 | E |
| Step 6 | 135 | 219 | 204 | m209 | | Step 6 | 91 | x | 158 | m210 |

| step | ID7 | ID10 | ID11 | output | | step | ID7 | ID10 | ID11 | output |
| --- | --- | --- | --- | --- | --- | --- | --- | --- | --- | --- |
| | input | | | | | | input | | | |
| Step 1 | 118 | 249 | 250 | m200 | | Step 1 | 117 | 253 | 248 | m200 |
| Step 2 | 123 | 235 | 252 | E | | Step 2 | 123 | 235 | 252 | E |
| Step 3 | 123 | 235 | 252 | E | | Step 3 | 123 | 235 | 252 | E |
| Step 4 | 114 | 251 | 243 | m200 | | Step 4 | 113 | 254 | 245 | m200 |
| Step 5 | 51 | 210 | x | m211 | | Step 5 | 69 | 130 | 170 | m212 |

| step | ID7 | ID10 | ID11 | output | | step | ID7 | ID10 | ID11 | output |
| --- | --- | --- | --- | --- | --- | --- | --- | --- | --- | --- |
| | input | | | | | | input | | | |
| Step 1 | 113 | 257 | 240 | m200 | | Step 1 | 112 | 250 | 249 | m200 |
| Step 2 | 123 | 235 | 252 | E | | Step 2 | 123 | 235 | 252 | E |
| Step 3 | 123 | 235 | 252 | E | | Step 3 | 123 | 235 | 252 | E |
| Step 4 | 115 | 258 | 237 | m200 | | Step 4 | 135 | 241 | 249 | E |
| Step 5 | 108 | 224 | 240 | E | | | | | | |
| Step 6 | 102 | 223 | 233 | TWO | | | | | | |

Verification of the decision-making algorithm for the identification of selected failures (step 1) starts at a failure-free state for each type of failure (the algorithm output is m200).

In step 2, the situation occurs, in which the decision-making algorithm evaluates as an unidentified failure (the algorithm output is E).

If a situation occurs as in step 3 in the trace table (the algorithm output is E), then the device remains in operation, as the time since the unknown failure identification is less than 45 s, i.e., changes in contact forces (CFs) can be induced by the dynamic nature of the device.

In step 4, the stabilization of vibrations is simulated until the algorithm result is m200, or a situation is simulated where the algorithm output is an unidentified failure (E) three times in a row; thus: E, E, E. Then, the operator must stop the pipe conveyor operation and visually inspect the cause of the failure in the idler housing.

In step 5, failures are simulated (missing roller at selected positions m2007 to m212, or absent material TWOMA0). If the failure type is clearly determined in this step (algorithm output is m208, m211, m212), then the simulation is finished, and the device is stopped. If the algorithm output is an unidentified failure (E), the simulation continues with the next step (step 6).

In step 6, the simulation sets unambiguous values for the contact forces (CFs) at measured positions, so that the output is one of the defined failures (the algorithm output is m207, m209, m210, or TWOMA0).

Based on the trace table (Table 11) and knowledge of the pipe conveyor, the failures were always identified sufficiently in advance, thanks to which, in practice, there would be no further damage to the pipe conveyor. The created algorithm can also be used for other types of failures, which are not considered in this paper due to the unavailability of specific measurements for contact forces (CFs). Then the algorithm output is E, which means stopping the operation of the pipe conveyor and a physical inspection of the idler housing by the operator.

Within this paper, we present a method using discrimination that was developed with the intention to digitally transform the identification of a failure in the idler housing. The principle of discrimination is that, based on different combinations of CF values at three positions, one of the properties is assigned to the device: failure-free state, missing roller including its position, or other failures. Compared to the results presented in [21–25], the solution is more complex and more innovative. However, this research did not examine a single method application to identify one type of damage. A result (algorithm) was obtained that will enable the implementation of digital transformation and the future identification of several types of damage.

The research on belt conveyors published so far only deals with estimating the parameters in regression models to determine the contact forces as a relation between tension forces and the conveyor belt's filling in different situations, for example, the asymmetry in tension forces [32] or in the conveyor belt's rotation [33].

This research succeeded in creating an algorithm, presented as a development diagram, that can identify a serious failure in a very short time. According to available information, such an approach has not been published yet; therefore, it can be considered innovative. With continuing research on the given topic, the mentioned method will be implemented into operational conditions and will contribute to increasing the safety and efficiency of continuous transport systems' operation.

## 5. Conclusions

Online monitoring of pipe conveyor operation with the use of continuous measurement is a demanding but highly effective means for identifying unfavorable conditions and ensuring operational reliability. In the process, the discrimination method can be used effectively.

The discrimination method was verified using simulated data (measured contact forces (CFs)), which characterized the properties of the used pipe conveyor test stand. The results confirm the suitability of the method for use in the practical assurance of a failure-free pipe conveyor operation.

In further research, the authors assume:

- Verification of the proposed discrimination method based on data obtained from real operations. For this, it will be necessary to set the frequency for scanning and evaluating the values of measured contact forces and the contact force (CF) variability in a failure-free operation.
- Verification of the possibility to identify other types of failures.
- Determining the impact of failures on other (adjacent) idler housings.
- Deriving a rule to determine on which idler housing, within the entire pipe conveyor route, the size of contact force (CF) needs to be monitored.

This presented research was carried out on a specific type of continuous transport system (pipe conveyor). Due to its specificity, the conclusions obtained cannot be considered generally valid for other types of continuous transport systems. Our results were confirmed using parameters such as tension force, contact forces, etc., which differ in individual types of continuous transport systems. Based on this, further research needs to be carried out focusing on other groups of continuous transport systems, aiming to create universally valid postulates with general validity and usability for the needs of digital transformation in intralogistics processes.

This presented research created basic prerequisites for digital transformation implementation. With its help, autonomous control algorithms can be created in the future, enabling automated control of continuous transport systems from the point of view of their standard operation, while at the same time identifying undesirable operating conditions to increase operational safety and reliability. The said solution will enhance efficient continuous transport system operation, and, thanks to digital transformation implementation, it will be possible to ensure that these intralogistics components meet the criteria of Industry 4.0. The created algorithm needs to be further refined and researched to achieve its universal application and subsequent use for other types of continuous transport systems.

**Author Contributions:** Conceptualization, V.M.; methodology, G.F. and V.M.; investigation, B.S. and D.K.; formal analysis, B.S. and G.F.; writing—original draft preparation, V.M. and G.F.; writing—review and editing, B.S. and P.M.; validation, P.M. and D.K. All authors have read and agreed to the published version of the manuscript.

**Funding:** This work is part of the projects VEGA 1/0488/23, VEGA 1/0101/22, VEGA 1/0264/21, VEGA 1/0600/20, KEGA 005TUKE-4/2022, KEGA 018TUKE-4/2022, and APVV-21-0195.

**Institutional Review Board Statement:** Not applicable.

**Informed Consent Statement:** Not applicable.

**Data Availability Statement:** Data sharing is not applicable to this article.

**Conflicts of Interest:** The authors declare no conflict of interest.

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
