# Peer review of "Identification of Selected Failures in a Pipe Conveyor’s Operation with the Use of the Discrimination Method Based on Continuous Measurement"

_applsci, doi:10.3390/app13126864_

Round 1

Reviewer 1 Report

The paper is of sufficient scientific interest in the concerned field and the subject of the paper is related to the aim and scope of the journal. The paper is well organized. The introduction section is informative. The manuscript is clear and concise. The references are adequate, satisfactory and given correctly. But there are some issues missing in this paper and hence, the manuscript needs minor revision based on the following comments:

1.      Is necessary to improve the abstract, indicating more precise and summarized data. The abstract should be much more concise. Some key results should be quantitatively shown.

2.      Flow of information has not been maintained in the literature review. The authors should have clearly indicated the objective of the present work with reference to the previous works. This would have substantially increased the importance of this research work. The authors should to explain better what are their innovative contribute?

3.      Figure 6 is not readable. The image quality and resolution should be improved.

4.      Discussion of results can be improved in the manuscript. It is very important to emphasize Theo points of agreement or disagreement between the results in the present work and the cited references in the manuscript however qualitative or quantitative may be.

5.      The conclusion  section should present the main findings of the study.

 Language needs to be refined throughout the manuscript. A lot of punctuation and grammatical mistakes are found in the current form of this manuscript.

Reviewer 2 Report

This research paper investigates the use of online diagnostics in monitoring the operational process of a pipe conveyor to identify potential failures during operation, such as a missing roller in the idler housing or absent material on the conveyor belt, using a discrimination method. The method, based on continuous measurement and evaluation of contact forces at specific rollers’ positions, was tested under 8 different regimes and validated using simulated data. The results confirmed the method's effectiveness in predicting failures sufficiently early to prevent further damage to the system. However, the following comments should be addressed before considering for publication.

1.       While the authors have extensively explained the concept of the pipe conveyor in the introduction, they have not introduced the methodology employed in the study within the same section. It is advisable that they incorporate details about the utilized methodology in the introduction section as well. Also, the motivation for this study should also be introduced in the introduction section.

2.       In section 2 line 113, the authors claimed that a specially developed test stand was used to perform all measurements of contact forces CF on the idler housing's rollers. Further details surrounding this test stand would be useful to provide additional context.

3.       The descriptions provided for cluster analysis and decision tree in sections 2.1 and 2.2 are quite succinct. It would be beneficial if the authors could expand on these methodologies, providing a more comprehensive explanation.

4.       In section 3 line 169, the authors state "In step 1 of identification of selected failures in the pipe conveyor’s operation, using the discrimination method (Figure 2)." Given the importance of the discrimination method to this study, more explicit information about the procedures depicted in Figure 2 would be valuable. For instance, after the dynamic nature of the transport system is implemented (step 4), why does the process proceed to the yellow rectangle, where the decision-making algorithm is created and tested? A more thorough explanation would enhance comprehension.

5.       Regarding the visual materials in the study, there are various colored cells within several tables; for instance, Table 5 features a yellow cell, while four cells in Table 6 are red. The significance of these colors needs to be clarified. Additionally, the figures lack clarity and would benefit from being presented in higher resolution, especially Figure 6, as its current quality makes it difficult for the reviewer to discern the details.

6.       The authors state that the discrimination method is effectively usable in line 405, yet there is no comparative analysis provided within the study to substantiate the effectiveness of this method relative to other available methods. A comparison would strengthen their claim.

7.       In conclusion, the authors need to acknowledge any limitations of the proposed method. This will help readers better understand the model's applicability and identify areas where further development and refinement may be needed. The reviewer suggests that the authors provide a clear discussion of any limitations to the proposed approach.
